# Scalable Hyperparameter Transfer Learning

**Valerio Perrone, Rodolphe Jenatton, Matthias Seeger, Cédric Archambeau**
Amazon
Berlin, Germany
{vperrone, jenatton, matthis, cedrica}@amazon.com

## Abstract

Bayesian optimization (BO) is a model-based approach for gradient-free black-box function optimization, such as hyperparameter optimization. Typically, BO relies on conventional Gaussian process (GP) regression, whose algorithmic complexity is cubic in the number of evaluations. As a result, GP-based BO cannot leverage large numbers of past function evaluations, for example, to warm-start related BO runs. We propose a multi-task adaptive Bayesian linear regression model for transfer learning in BO, whose complexity is linear in the function evaluations: one Bayesian linear regression model is associated to each black-box function optimization problem (or task), while transfer learning is achieved by coupling the models through a shared deep neural net. Experiments show that the neural net learns a representation suitable for warm-starting the black-box optimization problems and that BO runs can be accelerated when the target black-box function (e.g., validation loss) is learned together with other related signals (e.g., training loss). The proposed method was found to be at least one order of magnitude faster than competing methods recently published in the literature.

## 1 Introduction

Bayesian optimization (BO) is a well-established methodology to optimize expensive black-box functions [1]. It relies on a probabilistic model of an unknown target $f(\mathbf{x})$ one wishes to optimize and which is repeatedly queried until one runs out of budget (e.g., time). Queries consist in evaluations of $f$ at hyperparameter configurations $\mathbf{x}^1, \ldots, \mathbf{x}^n$ selected according to an explore-exploit trade-off criterion or *acquisition function* [2, 3, 1]. The hyperparameter configuration corresponding to the best query is then returned. One popular approach is to impose a Gaussian process (GP) prior over $f$ and, in light of the observed queries $f(\mathbf{x}^1), \ldots, f(\mathbf{x}^n)$, to compute the posterior GP. The GP model maintains a posterior mean function and a posterior variance function that are required when evaluating the acquisition function for each new query of $f$.

Despite their flexibility and ability to calibrate the predictive uncertainty, standard GPs scale cubically with the number of observations [4]. Hence, they cannot be applied in situations where $f$ has been or can be queried a very large number of times. A possible alternative is to consider sparse GPs, which scale linearly in the number of observations and quadratically in the number of inducing points [5, 6]. However, tractability requires the number of inducing points to be much smaller than the number of observations, resulting in a severe deterioration of the predictive performance in practice [7].

In this work, we aim to warm-start BO in the context of hyperparameter optimization (HPO). Our goal is to learn across related black-box optimization problems by transferring information between them, thus leveraging data from previous BO runs. For example, we warm-start the HPO of a given classifier when it is applied to a battery of reference data sets. Earlier work adopting this transfer learning perspective includes [8, 9]. To circumvent the scalability limitation of GPs and enable transfer learning in HPO at scale, we propose falling back to adaptive Bayesian linear regression (ABLR) [10], which scales linearly in the number of observations and cubically in the dimension of a

learned basis function expansion, hence the name *adaptive*. Our first contribution is to extend ABLR to the multi-task setting. Here, a *task* denotes a black-box function optimization, associated with its own Bayesian linear regression surrogate. These models share an underlying feedforward neural net (NN) that learns a shared basis expansion (or *representation*) from the HPO data. Our second contribution is to learn this representation *while* performing BO. This is achieved by integrating out the linear regression weights and learning the remaining parameters of each linear regression and NN weights by optimizing the overall log marginal likelihood. To this end, we leverage automatic differentiation operators recently contributed to MXNet [11, 12].

It is well-known that BLR can be seen as GP regression with a linear kernel, and the linear scaling with respect to the number of observations has long been established [13]. Most published work on HPO uses GPs with nonlinear kernels, as they provide more realistic predictive variances away from the observations [14, 15]. Our simpler and more scalable ABLR models are known to have weaknesses in that respect, which are shared by most other scalable GP approximations [7]. A notable exception is Deep Kernel Learning (DKL) [16]: a GP approximation with better predictive variance properties. ABLR can be viewed as a special case of DKL that is substantially simpler to implement and less costly to run, as it does not involve unrolling iterative linear solvers in a deep NN computation graph, nor specialized numerical interpolation code. While a sparse GP approximation was applied to standard BO in [17], we are not aware of previous work applying scalable GP approximations, including DKL, to transfer learning in HPO.

The paper is organized as follows. Section 2 summarizes the BO paradigm and relates our contributions to the state-of-the-art. Section 3 introduces our multi-task adaptive Bayesian linear regression, detailing inference in this model and discussing computational properties. We also explain how the model and its attractive computational properties can be exploited for efficient transfer learning. Section 4 presents experiments on simulated and real data, reporting favorable comparisons with existing alternatives when leveraging data across auxiliary tasks and signals. We conclude with a discussion of possible extensions in Section 5.

## 2   Background

Consider the problem of optimizing a black-box function $f(\mathbf{x}) : \mathcal{X} \to \mathbb{R}$ over a convex set $\mathcal{X} \subset \mathbb{R}$, namely a function whose analytical form and gradients are unavailable and that can only be queried through expensive and potentially noisy evaluations. For instance, suppose $f(\mathbf{x})$ is the test error associated to a deep neural network as a function of its hyper-parameters $\mathbf{x}$ (e.g., the number of layers, units, and type of activation functions). In this setting, each evaluation of $f(\mathbf{x})$ is typically very expensive as it requires training the neural network model.

BO is an efficient approach to find $\mathbf{x}_\star = \operatorname{argmin}_{\mathbf{x} \in \mathcal{X}} f(\mathbf{x})$. The idea is to place a surrogate model over the target black-box, and update it sequentially by querying $f(\mathbf{x})$ at new points that optimize an acquisition function, effectively trading off exploration and exploitation. Let $\mathcal{M}$ be a cheaper-to-evaluate surrogate model of $f(\mathbf{x})$, and $\mathcal{C}$ a set of evaluated candidates. The canonical BO loop iterates the following steps until some given budget, such as time, is exceeded:

1. Select a candidate $\mathbf{x}_{\text{new}} \in \mathcal{X}$ that optimizes a given acquisition function based on $\mathcal{M}$ and $\mathcal{C}$.
2. Collect an evaluation $y_{\text{new}}$ at $f(\mathbf{x}_{\text{new}})$.
3. Update $\mathcal{C} = \mathcal{C} \cup \{(\mathbf{x}_{\text{new}}, y_{\text{new}})\}$.
4. Update the surrogate model $\mathcal{M}$ based on $\mathcal{C}$.
5. Update the budget.

A variety of models $\mathcal{M}$ have been used to describe the black-box function $f(\mathbf{x})$, with GPs being a common choice. In the next subsection, we review a set of alternative models that have been proposed to either overcome the scalability limits of GPs or extend BO to optimize multiple related black-box functions.

### 2.1   Related work

Our work is most closely related to DNGO [18], where a scalable ABLR model is used for BO. However, this model is limited to a single task with many evaluations, while we aim to transfer

learning across multiple tasks. In addition, we do not use their two-step learning procedure. Namely, the authors first train the NN together with a final *deterministic* linear regression layer, relying on standard deep learning software. Then, they discard the final layer and replace it with a Bayesian linear regression model in order to drive BO. Our empirical results indicate that joint *Bayesian* learning of the ABLR parameters and the underlying NN parameters is beneficial, justifying the additional complexity in our implementation. Our procedure naturally extends to handling heterogeneous signals. Moreover, our comparisons with DNGO in Section 4 show that our approach runs significantly faster. More details on the relationship to DNGO are provided in Section 3.1 and 3.2.

Another related model is BOHAMIANN [15]. The authors propose using Bayesian NNs [19] to sample from the posterior over $f$ and add task-specific embeddings to the NN inputs to handle multi-task black-box optimization. While allowing for a principled treatment of uncertainties, fully Bayesian NNs are computationally expensive and their training can be sensitive to the stochastic gradient MCMC procedure used to handle the hyperparameters. Our model allows for simpler inference and is parameter free, making it more suitable for deployment at a scale in practice. As shown in Section 4, it runs much faster than BOHAMIANN.

Another line of related research does not rely on NNs. Feurer et al. [20] warm-start HPO with the best hyperparameters for the most similar black-box function, where similarity is measured by distance between the corresponding meta-data. Our multi-task ABLR model learns a useful shared feature basis *even in the absence* of task meta-data. It is able to draw information from *all* previous function evaluations, without having to restrict itself to the best solution from previous BO runs. This is similar in spirit to previous work [21, 22, 23], where the covariance matrix of a GP is designed to use the entire set of previous evaluations and capture black-box function similarities. Multi-task ABLR makes it possible to fully embrace this idea, as it can leverage orders of magnitude more observations than with a GP-based approach.

Finally, a number of models have been proposed specifically in the context of transfer learning for HPO. Schilling et al. [24] model the interaction between data sets and optimal hyperparameters explicitly with a factorized multilayer perceptron. Since this model cannot represent uncertainties, an ensemble of 100 multilayer perceptrons is trained to get predictive means and simulate variances. Golovin et al. [25] consider transfer learning in the particular setting where a sequence order (e.g., time) is assumed across the BO runs; we do not require this assumption. A different approach is taken by Wistuba et al. [26, 27]. In the former work, a meta-loss function is minimized to learn initial hyperparameter configurations. However, their method requires manually setting a kernel bandwidth to combine the predictive means of the past models, and an ad hoc procedure for the uncertainty which ignores the predictive variances of the past models. In the latter work, a two-stage surrogate model is considered: an independent GP is trained for each data set, after which kernel regression combines the GPs into an overall surrogate model for BO. The idea of using a mixture of GP experts and learning the weights of the ensemble is also proposed in Feurer et al. [28]. While the resulting models are able to exploit data set similarities, the cubic scaling makes GP-based approaches unfeasible with a large number of evaluations.

## 3  Multi-task Adaptive Bayesian Linear Regression

Consider $T$ tasks, which consist in the target black-box functions $\{f_t(\cdot)\}_{t=1}^T$ we would like to optimize and which are related in some way (e.g., the validation losses of a classification model learned on different data sets). We have evaluated $f_t(\cdot)$ $N_t$ times, resulting in the data $\mathcal{D}_t = \{(\mathbf{x}_t^n, y_t^n)\}_{n=1}^{N_t}$, also denoted by $\mathbf{X}_t \in \mathbb{R}^{N_t \times P}$ and $\mathbf{y}_t \in \mathbb{R}^{N_t}$ in stacked form. Our joint model for the responses $\mathbf{y}_t$ consists of two parts. First, we use a shared feature map $\phi_{\mathbf{z}}(\mathbf{x}) : \mathbb{R}^P \mapsto \mathbb{R}^D$. In our main use case, $\phi_{\mathbf{z}}(\mathbf{x})$ is a feedforward NN with $D$ output units, akin the model proposed in [29], and vector $\mathbf{z}$ collects all its weights and biases. We collect the features in matrices $\boldsymbol{\Phi}_t = \boldsymbol{\Phi}_{\mathbf{z}}(\mathbf{X}_t) = [\phi_{\mathbf{z}}(\mathbf{x}_t^n)]_n \in \mathbb{R}^{N_t \times D}$. Second, we employ separate Bayesian linear regression surrogates that share the feature map $\phi_{\mathbf{z}}(\mathbf{x})$ to model the black-box functions:

$$P(\mathbf{y}_t|\mathbf{w}_t, \mathbf{z}, \beta_t) = \mathcal{N}(\boldsymbol{\Phi}_t \mathbf{w}_t, \beta_t^{-1} \mathbf{I}_{N_t}), \quad P(\mathbf{w}_t|\alpha_t) = \mathcal{N}(\mathbf{0}, \alpha_t^{-1} \mathbf{I}_D),$$

where $\beta_t > 0$ and $\alpha_t > 0$ are precision (i.e., inverse variance) parameters. The model adapts to the scale and the noise level of the black-box function $f_t$ via $\beta_t$ and $\alpha_t$, while the underlying NN parametrized by a shared vector $\mathbf{z}$ learns a representation to transfer information between the black-box functions. Importantly, the weights $\mathbf{w}_t$ parametrizing the $t^{\text{th}}$ Bayesian linear regression

are treated as latent variables and integrated out, while the remaining parameters $\alpha_t, \beta_t$ and $\mathbf{z}$ are learned. The ABLR model can be seen as a NN whose final linear layers are Bayesian in the sense that their weights are integrated out rather than learned, or as a set of Bayesian linear regressions with a shared feature set learned by the NN. Note that Bayesian inference is analytically tractable and computationally efficient if restricted to the linear regression weights $\{\mathbf{w}_t\}_{t=1}^T$. Next, we provide expressions for the ABLR predictive probabilities and learning criterion. Detailed derivations can be found in the supplemental material.

### 3.1 Posterior Inference and Learning

Fixing the NN parameters and the precisions, the posterior distribution $P(\mathbf{w}_t|\mathcal{D}_t)$ over the linear regression weights are multivariate Gaussians, whose parameters can be computed analytically [10]. Moreover, if $\boldsymbol{\phi}_t^* = \boldsymbol{\phi}_{\mathbf{z}}(\mathbf{x}_t^*)$ is a new input for task $t$ and $f_t^* = \mathbf{w}_t^\top \boldsymbol{\phi}_t^*$ is the noise-free function value, the *predictive distribution* $P(f_t^*|\mathbf{x}_t^*, \mathcal{D}_t) = \int P(f_t^*|\mathbf{x}_t^*, \mathbf{w}_t)P(\mathbf{w}_t|\mathcal{D}_t)\, d\mathbf{w}_t$ is Gaussian as well. We show in the supplemental material that $P(f_t^*|\mathbf{x}_t^*, \mathcal{D}_t) = \mathcal{N}(\mu_t(\mathbf{x}_t^*), \sigma_t^2(\mathbf{x}_t^*))$, where

$$\mu_t(\mathbf{x}_t^*) = \frac{\beta_t}{\alpha_t}(\boldsymbol{\phi}_t^*)^\top \mathbf{K}_t^{-1}\boldsymbol{\Phi}_t^\top \mathbf{y}_t = \frac{\beta_t}{\alpha_t}\mathbf{e}_t^\top \mathbf{L}_t^{-1}\boldsymbol{\phi}_t^*, \quad \sigma_t^2(\mathbf{x}_t^*) = \frac{1}{\alpha_t}(\boldsymbol{\phi}_t^*)^\top \mathbf{K}_t^{-1}\boldsymbol{\phi}_t^* = \frac{1}{\alpha_t}\|\mathbf{L}_t^{-1}\boldsymbol{\phi}_t^*\|^2.$$

Here, $\mathbf{K}_t = \beta_t \alpha_t^{-1}\boldsymbol{\Phi}_t^\top \boldsymbol{\Phi}_t + \mathbf{I}_D$, and $\mathbf{L}_t$ is its Cholesky factor: $\mathbf{K}_t = \mathbf{L}_t\mathbf{L}_t^\top$. Moreover, $\mathbf{e}_t = \mathbf{L}_t^{-1}\boldsymbol{\Phi}_t^\top \mathbf{y}_t$. The predictive mean and the predictive variance drive the BO. Indeed, these are required to compute the acquisition function, which is instrumental to identify the most promising hyperparameters to evaluate (see [1] for a review and possible acquisition functions).

A key difference between our treatment of ABLR and DNGO is how the parameters $\{\alpha_t, \beta_t\}_{t=1}^T$ and $\mathbf{z}$ are learned. In DNGO, the NN weights $\mathbf{z}$ and the weights of the final layer $\mathbf{w}_1$ (they consider $T = 1$ only) are first learned by stochastic gradient descent. Next, $\mathbf{z}$ is fixed while $\mathbf{w}_1$ are discarded and estimated in subsequent BO rounds [18]. By contrast, we make no difference between BO and learning, integrating out the latent weights $\mathbf{w}_t$ in either case. The criterion we minimize is the negative log *marginal likelihood* of multi-task ABLR:

$$\rho\left(\mathbf{z}, \{\alpha_t, \beta_t\}_{t=1}^T\right) = -\sum_{t=1}^T \log P(\mathbf{y}_t|\mathbf{z}, \alpha_t, \beta_t),$$

where the marginal likelihood associated to task $t$ is given by $P(\mathbf{y}_t|\mathbf{z}, \alpha_t, \beta_t) = \mathcal{N}(\mathbf{y}_t|\mathbf{0}, \beta_t^{-1}\mathbf{I}_{N_t} + \alpha_t^{-1}\boldsymbol{\Phi}_t\boldsymbol{\Phi}_t^\top)$. As shown in the supplemental material, these quantities can also be expressed in terms of the Cholesky factor $\mathbf{L}_t$ of $\mathbf{K}_t$. Alternatively, when $N_t < D$, we can work with the Cholesky factor of $\mathbf{I}_{N_t} + \beta_t \alpha_t^{-1}\boldsymbol{\Phi}_t\boldsymbol{\Phi}_t^\top \in \mathbb{R}^{N_t \times N_t}$ instead. Hence, we can compute the learning criterion and its gradient in $\mathcal{O}(\sum_t \max(N_t, D)\min(N_t, D)^2)$.

In our model, each ABLR could be seen as a GP with shared linear kernel $\boldsymbol{\phi}_{\mathbf{z}}(\mathbf{x}_1)^\top \boldsymbol{\phi}_{\mathbf{z}}(\mathbf{x}_2)$, parameterized by $\mathbf{z}$. Minimizing $\rho$ is equivalent to learning these "kernel parameters" by empirical Bayes, as is routinely done for GPs [4]. By integrating out the linear regression weights, we induce the learned feature map $\boldsymbol{\phi}_{\mathbf{z}}(\mathbf{x})$ to provide a good representation for covariance and dependencies, not just for good point predictions. By contrast, DNGO jointly learns features and weights of a linear regression model, hoping that the former give rise to a useful covariance function. The results we present in Section 4 provide evidence for the superiority of empirical Bayes, at least in the multi-task setting.

### 3.2 Computational Implications

Our learning procedure comes with an additional complexity compared to the two-step approach of DNGO, where the model is trained using standard deep NN software and stochastic gradient descent (SGD) on mini-batches. While our learning criterion decouples as a sum over tasks, it does not decouple over the observations within a task: all $N_t$ observations for task $t$ form a single batch. If the number of tasks $T$ is moderate, our learning problem is best solved by batch optimization. In our experiments, L-BFGS [30] worked well.

Since Bayesian learning and optimization are grounded in the same principle, we can re-train all model parameters as part of BO, whenever new evaluations become available for a task. We adopt this approach in all our experiments and noted that L-BFGS re-converges in few steps because parameters change little with each new observation. In situations with a large number of tasks, we could run

BO on a task $t$ by only updating $(\alpha_t, \beta_t)$, not retraining the NN or updating the other parameters $\{\beta_{t'}, \alpha_{t'}\}_{t' \neq t}$. Full model retraining could then be done offline.

Our learning criterion cannot be expressed in standard deep NN software. Namely, the evaluation of the Bayesian linear regression negative log marginal likelihood requires computations such as $\mathbf{K}_t \mapsto \mathbf{L}_t$ (Cholesky decomposition) and $(\mathbf{L}_t, \mathbf{v}) \mapsto \mathbf{L}_t^{-1} \mathbf{v}$ (backsubstitution). These have to be available as auto-grad operators and should run on, both, CPU and GPU, so they can be first-class citizens in a computation graph. We implemented ABLR in MXNet [12], where a range of linear algebra operators have recently been contributed [11]. Given these operators, our implementation of ABLR is remarkably concise, and gradients required for model training and the minimization of the acquisition function are obtained automatically.

From a practical point of view, our approach has further advantages over DNGO. First, L-BFGS is simpler to use than SGD, as no parameters have to be tuned. This is all the more important in the context of BO, where our system has to work robustly on a wide range of problems without manual intervention. Second, we learn the parameters $\alpha_t$ and $\beta_t$ separately for each task by empirical Bayes [31], while such parameters would have to be manually tuned in DNGO. The critical importance of this point is highlighted in Section 4.4.

### 3.3 Transfer Learning Settings

In our experiments in Section 4, we consider a range of different use cases of BO with ABLR. The first use case we are interested in is HPO for a single machine learning model across different data sets. In this setting, a task consists in tuning the model on one of the data sets. Our goal is to warm-start HPO, so that a smaller number of evaluations are needed on a new data set, using the logs of previous HPO runs. The simplest approach is to learn a common feature basis $\phi_{\mathbf{z}}(\mathbf{x})$ across tasks, where each task is assigned to a separate marginal log likelihood term. If meta-features about the data are further available [20], we can collect them in a context vector $\mathbf{c}_t$, and use a map $\phi_{\mathbf{z}}(\mathbf{x}, \mathbf{c}_t)$ instead: the first part $\mathbf{x}$ of the input is variable, while the second part $\mathbf{c}_t$ is constant across data for a task.

Another use case is applying the ABLR model to a number of different signals (which play the role of tasks now). Here, we are interested in speeding up the optimization of one target function (e.g., validation loss), by leveraging a number of auxiliary signals (e.g., training cost, training loss considered at various epochs) which may come as a by-product, or are cheaper to evaluate. Since these different signals can differ widely in scale and noise level, the automatic learning of the scale parameter $\alpha_t$ and the noise $\beta_t$ is vitally important. Note that this set-up is different from a multi-objective scenario, such as the optimization of an average function over multiple tasks as described in [21]. Our set-up differs also from [23], since our primary task is fixed beforehand and we do not seek to identify the best source of information at each round.

## 4 Experimental Evaluation

The following subsections illustrate the benefits of multi-task ABLR in a variety of settings. In Sections 4.2 and 4.3, we evaluate its potential to transfer information between tasks defined by, respectively, synthetic data and OpenML data [32]. In Section 4.4, we investigate the transfer learning ability of ABLR in presence of multiple heterogeneous signals. In either setting, our goal is to accelerate BO by leveraging data from the related tasks and signals.

### 4.1 Experimental Set-up

We implemented multiple ABLR in GPyOpt [33], with a backend in MXNet [12], using recent linear algebra extensions [11]. The NN that learns the feature map $\phi_{\mathbf{z}}(\mathbf{x})$ is similar to the one used in [18]. It has three fully connected layers, each with 50 units and `tanh` activation function. Hence, 50 features are fed to the task-specific Bayesian linear regression models. We compare the NN set-up to random Fourier basis expansions [34], which have been successfully applied to BO [35, 36]. Specifically, let $\mathbf{U} \in \mathbb{R}^{D \times P}$ and $\mathbf{b} \in \mathbb{R}^D$ be such that $\mathbf{U} \sim \mathcal{N}(\mathbf{0}, \mathbf{I})$ and $\{b_j\}_{j=1}^D \sim \mathcal{U}([0, 2\pi])$. For a vector $\mathbf{x}$, the mapping is given by $\phi_{\mathbf{z}}(\mathbf{x}) = \sqrt{2/D} \cos(\sigma^{-1} \mathbf{U} \mathbf{x} + \mathbf{b})$, where $\sigma \in \mathbb{R}^+$ is the bandwidth of the approximated radial basis function kernel. We refer to this baseline as RKS ("random kitchen sink") in the remainder of the paper. It has only a single parameter $\sigma$ to optimize, which we learn

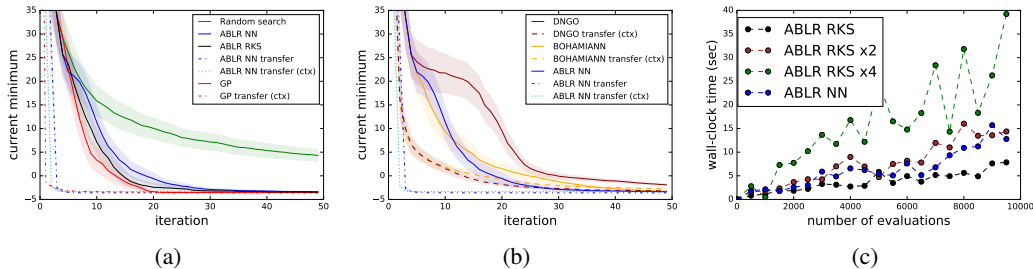

Figure 1: Results obtained on the parametrized quadratics (lower is better): (a) Transfer learning comparison of ABLR against baselines; (b) Transfer learning comparison of NN-based methods; (c) Run time of RKS- versus NN-based ABLR. See Section 4.2 for a discussion.

with the same L-BFGS code (see Section 3.2). We also compare ABLR-based BO to the standard GP-based BO, using GPyOpt. The GP has a Matérn-5/2 covariance kernel and automatic relevance determination hyperparameters, optimized by empirical Bayes [4].

In the experiments, we will consider models with and without transfer learning. All models without transfer are initialized according to GPyOpt default settings, that is, with a set of five evaluations picked at random. The models with transfer are initialized with one random evaluation from the target task. All BO experiments use the expected improvement acquisition function [2].

## 4.2 Transfer learning across parametrized quadratic functions

We first consider an artificial set-up with $T$ tasks, each given by a quadratic function defined on $\mathbb{R}^3$: $f_t(\mathbf{x}) = \frac{1}{2} a_{2,t} \|\mathbf{x}\|_2^2 + a_{1,t} \mathbf{1}^\top \mathbf{x} + a_{0,t}$, where $(a_{2,t}, a_{1,t}, a_{0,t})$ belongs to $[0.1, 10]^3$. This triplet can be thought of as the context associated to each task $t$. We generated $T = 30$ different tasks by drawing $(a_{2,t}, a_{1,t}, a_{0,t})$ uniformly at random, and evaluated multi-task ABLR against baseline methods and NN-based methods in a leave-one-task-out fashion. Specifically, we optimized each one of the 30 tasks after warm-starting the optimization with 10 observations drawn uniformly at random from each of the remaining 29 tasks. In other words, we have for each task $N_t = 10$ and warm-starting is therefore based on 290 evaluations. This set of observations is drawn once and taken the same for all other transfer learning methods. The results shown in Figure 1a and 1b are aggregates over 10 random repetitions of 30 leave-one-task-out runs.[1]

In Figure 1a, we compare single-task ABLR and standard GP driven HPO with their transfer learning counterparts. Transfer based on contextual information is denoted by `ctx`, using the context vector $\mathbf{c}_t = [a_{2,t}, a_{1,t}, a_{0,t}]^T$. We perform transfer learning in standard GPs by stacking all observations together and augmenting the input space with the corresponding contextual information [37]. Note that `GP transfer` learning uses a single marginal likelihood criterion over the data from all tasks, while `ABLR NN transfer` learning models the data from different tasks as conditionally independent. HPO converges to the minimum much faster for all transfer learning variants (leveraging data from 29 related tasks) than for the single-task ones. The single-task ABLR based on the RKS representation with $D = 100$ performed comparably to the one based on the NN representation with $D = 50$. The dimension $D = 100$ was picked after we investigated the computation time of ABLR-based HPO with learned NN features ($D = 50$) and with RKS features ($D \in \{50, 100, 200\}$) and found that the running times were similar (see Figure 1c). Figure 1a also shows that multi-task ABLR did not benefit much from the contextual information $\mathbf{c}_t$.

We further benchmarked single-task and transfer ABLR against the state-of-the-art NN-based approaches DNGO [18] and BOHAMIANN [15]. In contrast to GP-based methods, all these approaches scale linearly in the total number of evaluations. For their implementation we used the publicly available code `https://github.com/automl/RoBO`. We also used the recommended hyperparameters, which for BOHAMIANN were 2000 batches as burn-in followed by 2500 sampling steps. All NN architectures consisted of three fully connected layers, each with 50 units and `tanh` activation functions. Even though not considered in the original work [18], we extended DNGO to the transfer learning case in the same way as the GP baseline above, stacking all observations and augmenting the

input space with contextual information. Different to multi-task ABLR, a single marginal likelihood criterion is used over data from all tasks.[2] Results are shown in Figure 1b. The performance of single-task ABLR and BOHAMIANN is comparable (ABLR performs slightly better). DNGO and BOHAMIANN profit from transfer, yet less so than multi-task ABLR. Again, we note that the largest performance gain is realized *without* context input $\mathbf{c}_t$. This suggests that multi-task ABLR learns a useful joint representation through its shared feature map and better exploit similarities across tasks.

While the GP-based HPO with transfer slightly outperformed multi-task ABLR on the quadratic toy example, it does not scale to larger data sets, such as those considered in the next section. To make this more concrete, we measured the wall-clock time taken by HPO using GP and NN-based ABLR in a simple single-task setting. Our simulations showed that that GP-based HPO will not scale much beyond 2000 evaluations, which took approximately ten minutes, while ABLR-based HPO took only a few seconds (we provide curves in the supplemental material). These results indicate that GP-based HPO is problematic when considering transfer learning at scale.

Although all the considered NN-based algorithms scale linearly in $N$, with BOHAMIANN being slightly faster than DNGO (as observed in [15]), we found that our ABLR implementation requires much less computation time. More precisely, for the experiment in Figure 1b, the average time $\pm$ one standard deviation per BO iteration over 300 repeated runs, on CPU, amounted to about $1.7 \pm 0.10$ seconds for single-task ABLR and $28 \pm 0.15$ seconds for BOHAMIANN. In the following large-scale OpenML experiments, we report additional time comparisons with BOHAMIANN and DNGO.

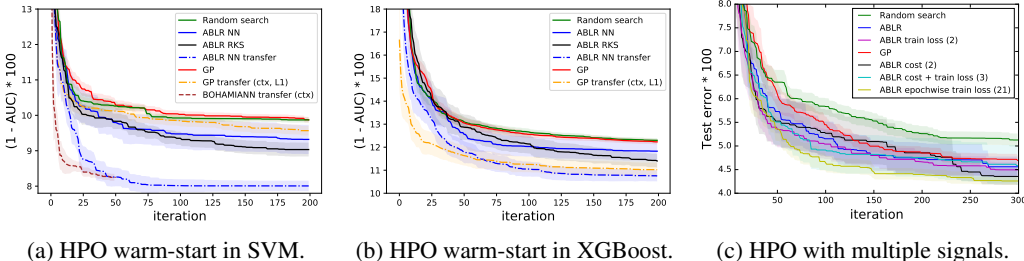

(a) HPO warm-start in SVM.    (b) HPO warm-start in XGBoost.    (c) HPO with multiple signals.

Figure 2: Transfer learning results obtained on OpenML and LIBSVM benchmarks (lower is better); see respectively Section 4.3 and Section 4.4 for a discussion.

### 4.3    Transfer learning across OpenML data sets

The ability to transfer knowledge across related tasks is particularly desirable in large-scale settings. Whenever runs from previous optimization tasks are available, these can be used to warm-start and potentially speed up the current optimization. We consider the OpenML platform [32], which contains a large number of evaluations for a wide range of machine learning algorithms (referred to as flows in OpenML) over different data sets. We focus on some of the most popular binary classification flows from OpenML, namely on a support vector machine (SVM, `flow_id` 5891) and extreme gradient boosting (XGBoost, `flow_id` 6767), and apply multi-task ABLR to optimize their hyperparameters. SVM comes with 4 and XGBoost with 10 hyperparameters. The parameters of the latter two exhibit conditional relationships, which we deal with by imputation [38]. More details on the OpenML set-up are given in the supplemental material. We filtered the $T = 30$ most evaluated data sets for each `flow_id`, which amounts to $\sum_t N_t \approx 6.5 \times 10^4$ ($N_t \in [1.087, 3.250]$) for SVM and $\sum_t N_t \approx 5.9 \times 10^5$ ($N_t \in [10.189, 48.615]$) for XGBoost. For these problems, the linear scaling of ABLR becomes almost mandatory. GP-based models cannot exploit all data even for a single task.

As previously, we apply a leave-one-task-out protocol, where each task stands for a data set. For the left-out task being optimized, say $t_0$, we use the surrogate modeling approach from [39] based on a random forest model. We compare single-task variants `GP`, `ABLR RKS`, and `ABLR NN` which use evaluations of task $t_0$ only, with `ABLR NN transfer` and `BOHAMIANN transfer (ctx)`, warm-started with the evaluations from the other 29 tasks, and `GP transfer (ctx, L1)`. In the last approach, we warm-start the GP model with 300 randomly drawn data points from the closest task in

terms of $\ell_1$ distance between contextual features (similar to [20]). In all OpenML experiments, we chose four contextual features: number of data points, number of features, class imbalance, and a landmark feature based on Naive Bayes. We did not use those based on ensemble methods to avoid potential information leakage about the targets. Note that the multi-task ABLR variant is not provided with context features.

Results are reported in Figures 2a and 2b, respectively for SVM and XGBoost, averaged over 10 random repetitions. The plots indicate that evaluation data from other data sets helps to speed up convergence on a new task. In particular, `ABLR NN transfer` is able to leverage such data by way of learning an efficient shared set of features. Further experiments are found in the supplemental material, also comparing the methods in terms of their mean ranking. We also provided the context vector $\mathbf{c}_t$ as input to multi-task ABLR and optimized the hyperparameters of a random forest model, but these settings did not lead to robust conclusions and further explorations are left for future work. Also note that the performance of `ABLR NN` ($D = 50$) and `ABLR RKS` ($D = 100$) is comparable. The real benefit of learning features by empirical Bayes is apparent in the multi-task scenario only.

We tried to compare to BOHAMIANN and DNGO in the same set-up, focusing on the SVM setting whose scale is smaller than that of XGBoost. As a reference, running `ABLR NN` and `ABLR NN transfer` in this setting took $1.2 \pm 0.2$ and $16.3 \pm 1.6$ seconds per BO iteration respectively. In contrast, BOHAMIANN needed $1335.7 \pm 236.5$ seconds per BO iteration, which was about 80 times more expensive. Therefore, we ran ABLR for 200 BO iterations, and limited BOHAMIANN to only 50 iterations due the high computational cost (see Figure 2a). Although BOHAMIANN was able to greatly speed up the optimization when warm-started and provided with contextual features, ABLR runs significantly faster and does not require dataset meta-data. As for DNGO, we did not succeed to run the method for more than a single iteration after which linear algebra errors related to the MCMC sampler cause the optimization to stop. This single iteration of DNGO took $15597.6 \pm 5833.5$ seconds, which is already 4 times as much as the total time of the 200 iterations of `ABLR NN transfer`. All our measurements are made on a `c4.2xlarge` AWS machine.

## 4.4 Tuning feedforward neural networks from heterogeneous signals

In a final experiment, we tune the parameters of feedforward NNs for binary classification. We use multi-task ABLR to simultaneously model $T$ signals associated to these feedforward NNs as outlined in Section 3.3. More specifically, we are interested in optimizing the validation error (i.e., the target signal) while modelling a range of auxiliary signals alongside (i.e., training error, training time, training error after $e$ epochs). Put differently, we use the multi-task nature of ABLR to model $T$ signals, learning a NN feature basis alongside a single HPO run. Importantly, the auxiliary signals come essentially for free, while most previous HPO algorithms do not seem to make use of them. Also note that different to the transfer learning settings above, we always evaluate all $T$ signals together, at the same input points $\mathbf{x}$. The fact that ABLR scales linearly in $T$ allows us to consider a large number of auxiliary signals (in contrast, multi-output GPs scales cubically in $T$). In our experiments, we tune four NN parameters: number of layers in $\{1, \ldots, 4\}$, number of units in $\{1, \ldots, 50\}$, $\ell_2$ regularization constant in $\{2^{-6}, 2^{-5}, \ldots, 2^3\}$, and learning rate of Adam [40] in $\{2^{-6}, 2^{-5}, \ldots, 2^{-1}\}$.

Results are provided in Figure 2c, averaged over 10 random repetitions and 5 data sets ($\{$`w1a`, `w8a`$\}$ [41], `sonar` [42], `phishing` [43, 42], `australian` [44, 42]) from LIBSVM [45]. The feedforward NN was trained for 200 iterations, each time on a batch of 200 samples. All variants consider the validation error as the signal of interest (and target for HPO). `ABLR train loss (2)` also uses the final value of the training loss, `ABLR cost (2)` the CPU training time, and `ABLR cost + train loss (3)` both. Finally, `ABLR cost + epochwise train loss (21)` uses the cost and the training error collected every 10 training iterations. In the model names, the number in parentheses denotes the number $T$ of signals modeled in ABLR. We can see that adding auxiliary signals to a HPO run driven by ABLR NN speeds up convergence. Note that this improvement comes from adding information which is available for free. We conjecture that adding auxiliary signals, related to the criterion of interest (e.g., gradient norms) would facilitate learning a useful feature basis by way of a feedforward NN, *even if* only one of these signals is the target of HPO. The ability to learn the parameters $\{\alpha_t, \beta_t\}$ per signal automatically is vital as some of the signals model with ABLR NN have different scales (e.g., validation error versus training time).

# 5 Conclusion

We introduced multi-task adaptive Bayesian linear regression (ABLR), a novel method for Bayesian optimization which scales linearly in the number of observations and is specifically designed for transfer learning in this context. Each task is modeled by a Bayesian linear regression layer on top of a shared feature map, learned jointly for all tasks with a deep neural net. Each Bayesian linear regression model comes with its own scale and noise parameters, which are learned together with the neural net parameters by empirical Bayes. When leveraging the auto-grad operators for the Cholesky decomposition [11], we found that training is at least as fast as the two-step heuristic recommended in [18].

We applied multi-task ABLR to two transfer learning problems in HPO. First, we investigated warm-starting HPO with synthetic optimization and meta-learning problems from OpenML. We demonstrated that multi-task ABLR converges considerably faster than GPs or other NN-based approaches, and scales to much larger sets of evaluations. We attribute the success of our method to its ability to learn a useful representation across tasks, even in the absence of meta-data. We speculate that this is due to the specific loss structure, which factorizes over the tasks. Multi-task ABLR further allows meta-data to be fed as context vectors to the underlying neural net, allowing the learned features to be task-specific without the need to design task distance metrics or requiring manual tuning. Second, we investigated multi-signal HPO for feedforward neural nets, showing that multi-task ABLR can leverage side-signals to speed up the optimization.

Several extensions are of interest. The Bayesian linear regression layers could be complemented by logistic regression layers in order to optimize binary signals or drive constrained HPO [46]. In a meta-learning context, we would have to further scale multi-task ABLR to a large number of tasks, a regime where batch learning by L-BFGS has to be replaced by stochastic optimization at the level of tasks. Finally, our joint Bayesian learning for deep NNs with a final Bayesian layer (which requires back-propagation through linear algebra operators such as Cholesky) can be applied to multi-task active learning or multi-label learning. Different to most other approximate Bayesian treatments of deep NNs [47, 48], we do not need random sampling or loosing variational bounding, but can fully leverage exact inference or tight approximation developed for generalized linear models.

## Footnotes

[1] The figures are reproduced in the supplement using a log scale to emphasize the gap between the curves.

[2] DNGO could also be used with a marginal likelihood criterion per task, but this would need substantial changes to their code. Also, the set of $\alpha_t$ and $\beta_t$ would have to be tuned for each task, which is impractical.

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
