[Supplementary Material]

# Supplementary material for the paper:
# Scalable Hyperparameter Transfer Learning

**Valerio Perrone, Rodolphe Jenatton, Matthias Seeger, Cédric Archambeau**
Amazon
Berlin, Germany
{vperrone, jenatton, matthis, cedrica}@amazon.com

## A Derivation of the Learning Criterion

In this section, we derive the learning criterion (negative log marginal likelihood) for Bayesian linear regression (BLR). Recall that our ABLR model is obtained by allocating one BLR model to each task $t$, where the features are shared and are given by a deep neural network.

The marginal likelihood for task $t$ is given by $P(\mathbf{y}_t|\mathbf{z}, \alpha_t, \beta_t) = \mathcal{N}(\mathbf{y}_t|\mathbf{0}, \beta_t^{-1}\mathbf{I}_{N_t} + \alpha_t^{-1}\boldsymbol{\Phi}_t\boldsymbol{\Phi}_t^{\top})$, and the learning criterion $\rho$ is its negative log. For now, assume that $N_t > D$ (more data points than features), and recall that $\boldsymbol{\Phi}_t \in \mathbb{R}^{N_t \times D}$. All parameters of the feature map are collected in $\mathbf{z}$. Up to an additive constant, we have

$$\rho\left(\mathbf{z}, \{\alpha_t, \beta_t\}_{t=1}^{T}\right) = \sum_{t=1}^{T} \frac{1}{2}(\log|\boldsymbol{\Sigma}_t| + \mathbf{y}_t^{\top}\boldsymbol{\Sigma}_t^{-1}\mathbf{y}_t), \quad \boldsymbol{\Sigma}_t := \alpha_t^{-1}\boldsymbol{\Phi}_t\boldsymbol{\Phi}_t^{\top} + \beta_t^{-1}\mathbf{I}_{N_t}. \quad (1)$$

We reformulate this expression in terms of the Cholesky decomposition of a $D \times D$ matrix. First, let $r_t = \beta_t/\alpha_t$ and consider the following identity:

$$\log|\mathbf{I} + r\boldsymbol{\Phi}\boldsymbol{\Phi}^{\top}| = \log|\mathbf{I} + r\boldsymbol{\Phi}^{\top}\boldsymbol{\Phi}|.$$

Then we have that

$$\log|\boldsymbol{\Sigma}_t| + N_t\log\beta_t = \log|\mathbf{I}_{N_t} + r_t\boldsymbol{\Phi}_t\boldsymbol{\Phi}_t^{\top}| = \log|\mathbf{K}_t|, \quad \mathbf{K}_t := \mathbf{I}_D + r_t\boldsymbol{\Phi}_t^{\top}\boldsymbol{\Phi}_t.$$

Decompose $\mathbf{K}_t$ in terms of the Cholesky factor $\mathbf{K}_t = \mathbf{L}_t\mathbf{L}_t^{\top}$ and use the Woodbury matrix identity

$$\boldsymbol{\Sigma}_t^{-1} = \beta_t(\mathbf{I}_{N_t} - r_t\boldsymbol{\Phi}_t\mathbf{K}_t^{-1}\boldsymbol{\Phi}_t^{\top}).$$

This gives

$$\mathbf{y}_t^{\top}\boldsymbol{\Sigma}_t^{-1}\mathbf{y}_t = \beta_t(\|\mathbf{y}_t\|^2 - r_t\|\mathbf{e}_t\|^2),$$

being $\mathbf{e}_t = \mathbf{L}_t^{-1}\boldsymbol{\Phi}_t^{\top}\mathbf{y}_t$. Plugging this into equation (1) yields the result:

$$\rho\left(\mathbf{z}, \{\alpha_t, \beta_t\}_{t=1}^{T}\right) = \sum_{t=1}^{T}\left[-\frac{N_t}{2}\log\beta_t + \frac{\beta_t}{2}\left(\|\mathbf{y}_t\|^2 - r_t\|\mathbf{e}_t\|^2\right) + \sum_{i=1}^{D}\log([\mathbf{L}_t]_{ii})\right], \quad r_t = \frac{\beta_t}{\alpha_t}.$$

As detailed in [1], $\rho$ can succinctly be expressed as an MXNet computation graph, using linear algebra operators under `mxnet.linalg`.

If $\phi_t^* = \phi_\mathbf{z}(\mathbf{x}_t^*)$ is the feature vector for a test point, we are interested in the predictive mean and variance of the noise-free function value $f_t^* = \mathbf{w}_t^{\top}\phi_t^*$: these quantities configure the acquisition function, used to make decisions on where to next evaluate the latent function. The posterior distribution of the weight vector $\mathbf{w}_t$ is Gaussian, with mean and covariance given by

$$\mathrm{E}[\mathbf{w}_t|\mathcal{D}_t] = r_t\mathbf{K}_t^{-1}\boldsymbol{\Phi}_t^{\top}\mathbf{y}_t = r_t\mathbf{L}_t^{-\top}\mathbf{e}_t, \quad \mathrm{Cov}[\mathbf{w}_t|\mathcal{D}_t] = (\alpha_t\mathbf{K}_t)^{-1}.$$

Figure 1: HPO warm-start in SVM.

Figure 2: HPO warm-start in XGBoost.

Then,

$$\mu_t = \mathrm{E}[f_t^*|\mathcal{D}_t] = (\boldsymbol{\phi}_t^*)^\top \mathrm{E}[\mathbf{w}_t|\mathcal{D}_t] = r_t \mathbf{e}_t^\top \mathbf{L}_t^{-1} \boldsymbol{\phi}_t^*,$$
$$\sigma_t^2 = \mathrm{Var}[f_t^*|\mathcal{D}_t] = (\boldsymbol{\phi}_t^*)^\top \mathrm{Cov}[\mathbf{w}_t|\mathcal{D}_t]\boldsymbol{\phi}_t^* = \alpha_t^{-1}\|\mathbf{L}_t^{-1}\boldsymbol{\phi}_t^*\|^2.$$

While the case $N_t > D$ is more prevalent in practice, we also encounter the situation $N_t \leq D$ (less data points than features). Some tasks may have few evaluations only, and we still like to use them. In particular, the target task of interest in HPO will start with very few initial evaluations. We found that if $N_t \leq D$, the following different expression for $\rho$ should be used both for reasons of computational efficiency and numerical robustness. First,

$$\boldsymbol{\Sigma}_t = \beta_t^{-1}\left(\mathbf{I}_{N_t} + r_t\boldsymbol{\Phi}_t\boldsymbol{\Phi}_t^\top\right) = \beta_t^{-1}\mathbf{E}_t\mathbf{E}_t^\top,$$

where $\mathbf{E}_t \in \mathbb{R}^{N_t \times N_t}$ is the Cholesky factor of $\mathbf{I}_{N_t} + r_t\boldsymbol{\Phi}_t\boldsymbol{\Phi}_t^\top$. Plugging this into (1) results in

$$\rho\left(\mathbf{z}, \{\alpha_t, \beta_t\}_{t=1}^T\right) = \sum_{t=1}^T \left[ -\frac{N_t}{2}\log\beta_t + \frac{\beta_t}{2}\|\mathbf{E}_t^{-1}\mathbf{y}_t\|^2 + \sum_{i=1}^{N_t}\log([\mathbf{E}_t]_{ii}) \right].$$

For the predictive mean and variance, we use the Woodbury formula to obtain

$$\mathbf{K}_t^{-1} = \mathbf{I}_D - r_t\boldsymbol{\Phi}_t^\top\mathbf{E}_t^{-\top}\mathbf{E}_t^{-1}\boldsymbol{\Phi}_t.$$

Plugging this into the equation above, some algebra gives

$$\mathrm{E}[\mathbf{w}_t|\mathcal{D}_t] = r_t\boldsymbol{\Phi}_t^\top(\mathbf{E}_t\mathbf{E}_t^\top)^{-1}\mathbf{y}_t = r_t\boldsymbol{\Phi}_t^\top\mathbf{E}_t^{-\top}\mathbf{E}_t^{-1}\mathbf{y}_t.$$

Therefore,

$$\mu_t = \mathrm{E}[f_t^*|\mathcal{D}_t] = (\boldsymbol{\phi}_t^*)^\top\mathrm{E}[\mathbf{w}_t|\mathcal{D}_t] = r_t(\mathbf{E}_t^{-1}\mathbf{y}_t)^\top\mathbf{E}_t^{-1}\boldsymbol{\Phi}_t\boldsymbol{\phi}_t^*,$$
$$\sigma_t^2 = \mathrm{Var}[f_t^*|\mathcal{D}_t] = (\boldsymbol{\phi}_t^*)^\top\mathrm{Cov}[\mathbf{w}_t|\mathcal{D}_t]\boldsymbol{\phi}_t^* = \alpha_t^{-1}\left(\|\mathbf{y}_t\|^2 - r_t\|\mathbf{E}_t^{-1}\boldsymbol{\Phi}_t\boldsymbol{\phi}_t^*\|^2\right).$$

We implement both variants ($N_t > D$ and $N_t \leq D$), and for each $t$, use the variant which applies.

## B  Additional experiments and OpenML set-up

In the OpenML [2] experiments we considered the optimization of the hyperparameters of the following.

- Support vector machine (SVM, `flow_id` 5891),
- Extreme gradient boosting (XGBoost, `flow_id` 6767).

Figure 3: Rankings, SVM.

Figure 4: Rankings, XGBOOST.

Figure 5: ABLR vs. baselines, quadratic functions (log scale)

Figure 6: Comparison of NN-based methods, quadratic functions (log scale).

## B.1 Support vector machine

The SVM tuning task consisted of the following 4 hyperparameters:

- cost (float, min: 0.000986, max: 998.492437),
- degree (int, min: 2.0, max: 5.0),
- gamma (float, min: 0.000988, max: 913.373845),
- kernel (string, [linear, polynomial, radial, sigmoid]).

This tuning task exhibits conditional relationships with respect to the choice of the kernel.[1]

For this `flow_id`, we considered the 30 most evaluated data sets whose `task_ids` are: 10101, 145878, 146064, 14951, 34536, 34537, 3485, 3492, 3493, 3494, 37, 3889, 3891, 3899, 3902, 3903, 3913, 3918, 3950, 6566, 9889, 9914, 9946, 9952, 9967, 9971, 9976, 9978, 9980, 9983.

## B.2 XGBoost

The XGBoost tuning task consisted of 10 hyperparameters:

- alpha (float, min: 0.000985, max: 1009.209690),
- booster (string, ['gbtree', 'gblinear']),
- colsample_bylevel (float, min: 0.046776, max: 0.998424),
- colsample_bytree (float, min: 0.062528, max: 0.999640),
- eta (float, min: 0.000979, max: 0.995686),

Figure 7: Data embeddings, SVM.

Figure 8: Data embeddings, XGBoost.

Figure 9: Embedding rankings, SVM.

Figure 10: Embedding rankings, XGBoost.

- lambda (float, min: 0.000978, max: 999.020893)
- max_depth (int, min: 1, max: 15),
- min_child_weight (float, min: 1.012169, max: 127.041806),
- nrounds (int, min: 3, max: 5000),
- subsample (float, min: 0.100215, max: 0.999830).

This tuning task exhibits conditional relationships with respect to the choice of the booster.[2]

For this `flow_id`, we considered the 30 most evaluated data sets whose `task_ids` are: 10093, 10101, 125923, 145847, 145857, 145862, 145872, 145878, 145953, 145972, 145976, 145979, 146064, 14951, 31, 3485, 3492, 3493, 37, 3896, 3903, 3913, 3917, 3918, 3, 49, 9914, 9946, 9952, 9967.

### B.3 Further details and experiments

For the OpenML experiments in the supplement we used a surrogate modeling approach [3] based on nearest neighbor. The results were averaged in a leave-one-dataset-out scheme, i.e., the model hyperparameters were optimized on each of the 30 available `task_ids` using each time the evaluations from the remaining `task_ids` to warm start the optimization.

A way to warm start GPs consists of collecting observations from the related tasks and augmenting them with contextual information. While speeding up the optimization, this approach does not scale well with the number of available evaluations. We explored different ways to circumvent this and achieve transfer learning with GPs in the large-scale OpenML experiments. In particular, we focused on two subsampling logics, namely `GP transfer (ctx, L1)` and `GP transfer (ctx, subsample)`. The former warm starts the BO with 300 points from the closest task, computed in terms of the L1 distance of the meta-data [4], while the latter uses 10 points for each of the 29 related

tasks (for a total of 290 points). Recall that this subsampling schemes are necessary to guarantee a feasible computational time for GPs. The results in Figures 1 and 2 indicate that the two approaches both outperform plain GP and achieve similar performance. At the same time, ABLR with transfer tends to converge to a better optimum, showing the benefits of the multi-loss structure and the advantages of scaling up to a larger number of observations for warm start.

We also compared against `BOHAMIANN transfer (ctx)` in the large-scale SVM experiment (Figure 1). As for the GP, we consider a version of this model where inputs are augmented with contextual vectors to achieve transfer across tasks. Due to its computational burden, we were able to run BOHAMIANN for 50 iterations and 3 independent replications of the leave-one-dataset-out scheme. While transferring knowledge across tasks proves to be beneficial for both BOHAMIANN and ABLR, the former suffers from a significantly larger computational overhead and relies on the availability of meta-data.

To better account for the heterogeneity across tasks, we also considered the mean ranking as an alternative metric. Specifically, we computed the relative ranking of the models for each target task separately, after which we averaged the rankings over all tasks. The results are given in Figures 3 and 4, where the shaded areas represent one standard error around the mean ranking. Warm-starting the BO helps converge faster and to a better optimum, with transfer ABLR tending to outperform all alternative approaches.

Finally, we provide log-scale plots of the quadratic function optimization results presented in the main text, where ABLR is compared against a set of baselines. More precisely, the $y$-axis shows $\log(y + 4)$ instead of $y$, the best minimum found up to each iteration. Figures 5 and 6 confirm the benefits of ABLR transfer, which converges considerably faster and to a better optimum compared to alternative approaches.

## B.4 Learning Task Embeddings

Instead of feeding context vectors $\mathbf{c}_t$ to the shared NN, we can use ABLR to *learn task embeddings*. This is of interest in a scenario where different ML methods are applied to a common (or at least overlapping) set of data sets. To this end, we use the NN form $\phi_{\mathbf{z}}(\mathbf{x}, \mathbf{c}_t)$, together with $\mathbf{c}_t = \mathbf{E}\boldsymbol{\delta}_t$. Here, $\mathbf{E} = [\mathbf{e}_t] \in \mathbb{R}^{k \times T}$ is a matrix of $T$ embedding vectors, and $\boldsymbol{\delta}_t \in \{0, 1\}^T$ is the one-hot vector for task $t \in \{1, \ldots, T\}$ (similar to the setting of [5]). The embedding matrix $\mathbf{E}$ becomes part of the overall parameter vector $\mathbf{z}$, trained by empirical Bayes. In some preliminary experiments, we use ABLR NN to transfer across *two* dimensions. We consider a Random Forest model (RF, `flow_id` 6794) and operate in two stages. First, for each of the three methods (RF, SVM, XGBoost) we learn NN features and task embeddings together over their respective $T = 30$ data sets. Second, we fix the task embeddings learned for one method and use them for another. In this second phase, we fix $\mathbf{E}$, but retrain the NN parameters. In our experiments, we fix the number $k$ of contextual features to four (thus matching the dimension of the four OpenML meta-features used). The 30 data sets we transfer between for each method are not the same, but have substantial overlap. SVM used the embeddings learned for XGBoost (15 data sets in common), and XGBoost used the embeddings learned for RF. The pairing was chosen to maximize the number of data sets in common. Results for this two-stage experiment are given in Figures 7 and 8. The learned embeddings are competitive with the context vectors provided from human-crafted OpenML meta-features, but more work would be needed to establish the usefulness of learned or hand-designed context vectors. We also applied the ranking metric to explore the usefulness of the task embeddings learned by ABLR (Figures 9 and 10), confirming that ABLR can infer task embeddings that perform similarly or better than human-crafted meta-data.

## Footnotes

[1]For details, we refer the interested readers to the API from `www.rdocumentation.org/packages/e1071/versions/1.6-8/topics/svm`

[2]For details, we refer the interested readers to the API from www.rdocumentation.org/packages/xgboost/versions/0.6-4