[Reviews · NeurIPS 2018]

Reviewer 1



This paper proposes a novel Bayesian Optimization approach that is able to do transfer learning across tasks while remaining scalable. Originality: This is very original work. Bayesian Optimization can work with any probabilistic regression algorithm, so the use of Bayesian linear regression to make it more scalable is well-known, as are its limitations (e.g. it doesn’t extrapolate well). The main novelty here lies in the extension to multi-task learning, which allows it to benefit from prior evaluations on previous tasks. When such evaluations are available, this can provide a significant advantage. The really clever part, however, is to use a neural net to learn a shared representation of the HPO space, and to learn this representation while performing Bayesian Optimization. Sections 1 and 2 provide a nice overview of related work, which seems quite complete to me. One very recent related paper that was not mentioned (maybe the authors were unaware of it) is the following arXiv paper: https://arxiv.org/abs/1802.02219 It does transfer learning by storing surrogate models of past tasks, and learns how to weight them in an ensemble during the Bayesian Optimization. One additional paper that comes to mind (since it also does transfer learning to warm-start BayesOpt) is the following: Wistuba, M. and Schilling, N. and Schmidt-Thieme, L. Learning hyperparameter optimization initializations, DSAA 2015 Clarity: The paper is very well written. It does require solid knowledge of Gaussian Processes to be fully understandable, but otherwise the method and experiments are explained concisely but clearly. Note: sometimes the explanations are a bit too concise, maybe because of page limits. For instance: L342: “When leveraging the auto-grad operators for the Cholesky” -> Cholesky decomposition. What is not entirely clear from the paper is how the method learns similarity between tasks/black box functions. Previous work always aims to learn how similar other tasks are, so it can transfer from the right tasks. From my understanding, it seems that the neural net does something similar in that the shared representation captures black-box function similarities. However, I didn’t find this explained very clearly. The results on 30 different OpenML datasets do seem to indicate that it learns similarity. Can this method also tell how similar a new task is to previous tasks, or actually, how similar a its black box function is to previous black box functions? Significance: This is a very significant contribution. Scalable transfer learning is somewhat of a holy grail in Bayesian optimization, so this paper will very likely have a significant impact, and within the right setting it will likely see significant adoption. Within the scope of the presented experiments, the method seems to work well. Quality: This is technically a very strong paper. The experiments are nice, although also limited in several regards: First, the current experiments are a bit limited in that they explore the hyperparameter spaces of single algorithms/functions. These spaces are typically still smallish (3-10 hyperparameters). It would be interesting to see it used on much larger hyperparameter spaces. Other related work, e.g. auto-sklearn with warm-starting, does operate on much larger hyperparameter spaces (e.g. entire pipelines containing a wide range of algorithms and all their hyperparameters). Second, the evaluation on OpenML datasets compares only to random search and GPs with transfer. It does mention that DNGO and BOHAMIANN were too slow or too unstable. However, it is not clear why other work that uses warm-starting or transfer learning is not compared against, even though this should be relatively easy on OpenML datasets? It would be good to at least comment on these limitations, or better, add the comparisons. Update: I grew more worried after reading the author responses. The question on comparing with other scalable HPO techniques was not answered directly. Why not compare against other established and robust and scalable AutoML methods, such as AutoSKLearn/SMAC? You could run that against the same surrogate benchmark? I agree that the methods that are currently compared against are in a way more related, but they also seem not scalable enough. One could argue that the method is interesting enough in itself and that the community can run further comparisons, but sadly there is no code available and the authors did not respond that it will be available after publication. This worries me.

Reviewer 2



The authors describe a large-scale transfer learning approach for tuning the hyperparameters of a machine learning model. They describe an adaptive Bayesian linear regression model. They attach one Bayesian linear regression model for each task and the information is combined by coupling the models through a deep neural net. They show extensive experimental results to validate the claims as well as good comparisons with the state-of-the-art. The paper is very well written and was a pleasure to read. It is highly significant to the field of autoML as it greatly reduces the time in obtaining optimal hyperparameters. The paper is technically correct and I would vote for a marginal accept. My detailed comments are given below. 1. My one concern is that there is a huge similarity with Deep Kernel Learning (DKL) which the authors acknowledge but they claim that "it is substantially simpler to implement and less costly to run". However, unless the code is shared this is a tough argument. Moreover, the DKL code is openly available and it would have been nicer to see a comparison between the two setups, especially using the KISS-GP approach for scalability. 2. An algorithm section seems to be lacking. Unless the reader is very familiar with the previous literature it might be a bit tough to follow. There are several mentions of BO and Acquisition Function but for someone who is unfamiliar with its use, it will very difficult to tie things together. 3. I really liked the extensive experiments section as it highlights the major impact of the paper. The only thing lacking is a comparison with DKL to portray the simplicity / less cost of this algorithm. Update: Although the authors gave a description of DKL and its drawbacks, the code was not made available and as a community, it might be tough to reproduce such results.

Reviewer 3



This work presents a method for transfer learning in hyperparameter optimization. The transfer learning takes the form of a feature mapping that is shared by all tasks (datasets), upon which a Bayesian linear regression model is then superposed. The main contribution of the paper is to propose an end-to-end learning algorithm able to jointly tune the underlying neural network features shared by all tasks. Some clever Bayesian parameterization allows to tune the representation jointly while integrating out the weights for each independent task. Quality: the quality of this paper is high, although the experiments were a bit underwhelming. The paper is lacking some clarification or justification on the use of a surrogate model for evaluation. Clarity: I found almost no errors or imprecisions in the presentation of the method. Originality: the work is novel and builds upon previous works in the literature. The ideas are straightforward and that is what makes them great. Significance: I think this work has its place in the literature on AutoML. Once people start sharing representations learned on thousands of problems, the process of optimizing a new algorithm's hyperparameter will be straightforward and quick. It does not suffer from the drawbacks of many other works on transfer learning in hyperparameter optimization (limited number of possible algorithm configurations, can only be used for warm-starting optimizations, etc.). As stated above, the experimental section of the work does hinder its significance. More comments on this below. General comments and questions: I find it very interesting the ABLR-NN with transfer learning was able to perform about as good with or without the context information, which was very strong information. On the OpenML experiments, to be clear, you didn't train any model on the OpenML datasets for the transfer part? What about for the final evaluation? From reading the supplementary material (line 65), I seem to understand that a surrogate model trained on the OpenML observations was used rather than training models? Am I correct in my assessment? Relying on a surrogate model for performance measurement, how can we be sure of the validity of the results? What if a model was sampled in a under-represented region of the hyperparameter space (esp. true for the 10-D space of XGBoost)? I think this should be explicited and justified in the paper, not pushed back to the supplementary material. Right now it feels a bit misleading. I think some follow-up experiments with a real evaluation of the performance by training and testing classifiers would make for a stronger case. Also, why use a nearest neighbor? From the experiments in Eggensperger et al. (2012), it was clearly the worst of all surrogate models. The performance of the models in Figure 2b) does not seem to have plateaued at iteration 200, they seem to be on a steady downward slope. Could this be a side effect of the use of a surrogate model for evaluation? In my experience, tree ensemble models have relatively easy to tune hyperparameters and do not benefit greatly from longer hyperparameter optimization budgets. line 298: "[running ... in this] setting take" should be "[running ... in this] setting takeS" Due to the potential issues with the use of a surrogate model, I am giving this paper a weak accept. I believe the experiments are still valid, but they are made weaker by this in my opinion (esp. when we can see artifacts of the use of such surrogate model, e.g. Figure 2b). It is possible that I misinterpreted some elements, in that case please enlighten me (and make the paper more clear on the matter). UPDATE: after reading author feedback, I still think the use of a surrogate model is strong limiting factor of the experimental study. This work would benefit strongly from doing real-world experiments (or at least doing part of the evaluation in the classical method).